# Research on the development and differentiation of the physical literacy scale for Chinese college students

**Rongjing Ni**, **Ying Yu***

Physical Education Institute, Liaoning Normal University, Dalian, China

* yuying640109@163.com

## Abstract

### Objectives

To understand the current status and characteristics of physical literacy among Chinese college students, develop a measurement scale suitable for this population, and provide an appropriate assessment tool for related research.

### Methods

The Delphi method and a questionnaire survey were initially employed to screen and evaluate the reliability and validity of 53 preliminary items. Subsequently, the finalized measurement tool, which had been assessed for reliability, was administered to 3,077 Chinese college students via a questionnaire survey to identify population variation characteristics.

### Results

Following a systematic screening process, the reliability coefficients for the four dimensions ranged from 0.731 to 0.821, and a reliable measurement tool for assessing physical literacy among Chinese college students was established. The final scale comprises four dimensions—physical activity capacity, physical activity behavior, physical activity cognition, and emotional experience—with a total of 32 items. The survey revealed significant differences in physical literacy among Chinese college students across the following dimensions: gender ($t = 24.914$, $P < 0.001$), household registration ($t = 11.464$, $P < 0.001$), grade ($F = 30.811$, $P < 0.001$), university category ($F = 17.305$, $P < 0.001$), and university district ($F = 4.957$, $P < 0.01$). Furthermore, significant differences were observed to varying degrees across all four dimensions—physical activity capacity, physical activity behavior, physical activity cognition, and emotional experience.

**Data availability statement:** All relevant data are within the manuscript and its Supporting Information files.

**Funding:** The author(s) received no specific funding for this work.

**Competing interests:** The authors have declared that no competing interests exist.

## Conclusion

The developed Physical Literacy Scale for Chinese College Students is suitable for research targeting this population. The survey revealed significant differences in physical literacy levels among Chinese college students based on gender, grade, household registration, university category, and university district.

## Introduction

In recent years, an increasing number of scholars have recognized that physical activity not only serves as a fundamental tool for human productive activities and the maintenance of normal life functions but also constitutes a crucial component in safeguarding physical and mental health [1–3]. To systematically understand the physical literacy of Chinese college students, it is essential to first grasp its structural characteristics and develop a scientifically standardized assessment tool. Existing research in related fields indicates that the dimensional classification and measurement methods for physical literacy vary across different populations. Although physical literacy is considered a lifelong indicator, previous studies have revealed structural differences in physical literacy across various life stages, regions, and other demographic groups [4–6]. The International Physical Literacy Alliance (IPLA) also emphasizes that each nation can define compatible physical literacy based on its unique cultural context and background. Currently, China lacks a fully developed measurement tool for assessing physical literacy among college students, which hinders related research. Therefore, to refine the measurement methods for college students' physical literacy, this section focuses specifically on this population to further understand the structural characteristics of physical literacy among Chinese college students and to develop measurement tools suitable for advancing subsequent research. On this basis, the developed College Student Physical Literacy Scale will be used to systematically explore population differences in physical literacy and its internal structure. The objective is to construct an evidence-based, multidimensional, and multilevel differentiated theoretical framework for promoting physical literacy, thereby effectively enhancing the overall level of physical literacy and promoting equitable health development among college students.

Gender differences represent a key factor examined across various studies. Numerous investigations reveal that boys exhibit higher physical literacy development levels than girls as early as childhood [7]. Among college students, females consistently demonstrate lower participation rates in sports activities, reduced physical activity volume and intensity, and longer sedentary periods compared to males—a finding repeatedly validated [8–10], and the reciprocal relationship between physical literacy and physical activity has also been clearly established [11]. Therefore, based on previous research experience, gender may be a significant factor contributing to the marked differences in physical literacy among Chinese college students.

Urban-rural disparities have consistently been a discussed element in physical activity-related research. For instance, a survey on physical activity among Chinese

children and adolescents revealed that weekly physical activity attainment rates were higher among urban children and adolescents than their rural counterparts across all regions [12]. Physical literacy reflects an individual's knowledge, skills, and abilities in physical activity, which is linked to regional differences in educational attainment and sports resources. For instance, disparities in school sports facilities, equipment, and overall campus environments between urban and rural areas impact the effectiveness of school-based physical activity promotion [13]. Consequently, students in rural areas tend to exhibit relatively lower levels of physical activity knowledge, cognition, and skills.

Scholars maintain that physical education remains a crucial means to promote physical activity among student populations. Physical education can enhance students' physical literacy through diverse sports activities [14,15]. Under China's prevalent physical education model, freshmen and sophomores receive standardized physical education courses, while juniors and seniors have fewer opportunities for sports-related courses, making them more susceptible to declining physical literacy. Moreover, higher grades are linked to reduced physical activity and motivation due to academic and career pressures, leading to lower physical literacy.

Environmental factors significantly influence college students' physical literacy. Adequate campus facilities, faculty support, sports events, and clubs enhance students' physical activity and literacy. Differences in institutional types and regional economic conditions—such as comprehensive vs. specialized universities, and eastern vs. western regions—may lead to varying levels of physical literacy among students.

However, contrary to the aforementioned findings, studies on Chinese student samples have also revealed that, although the overall level of physical literacy among Chinese college students is alarmingly low, no significant differences exist across gender, age, or region of origin [16]. This discrepancy warrants further investigation in the present study. Based on a preliminary literature analysis, the following hypothesis is proposed: college students' physical literacy and its dimensions vary according to gender, household registration, grade, university category, and university district.

## Methods

Preliminary theoretical analysis and research on college students' physical literacy have provided an initial understanding of its connotation and characteristics. Although measurement tools for assessing physical literacy among Chinese college students require further refinement, the evaluation indicator system for physical literacy is relatively well-established. Considering the characteristics of the university student population, the formulation of items and dimensions drew upon the Physical Literacy Evaluation Index System Part 1: Adults and Part 2: Adolescents. It also integrated the requirements of China's higher education physical health curriculum and referenced the National Student Physical Fitness Standards. This approach aimed to ensure the scientific rigor and comprehensiveness of the preliminary items. Currently, within the Physical Literacy Evaluation Index System published by major Chinese institutions, there is broad consensus on the primary dimensions of physical literacy, and these have been extensively validated. These dimensions mainly include physical activity capacity, physical activity behavior, physical activity cognition, and emotional experience. Accordingly, the item pool was primarily structured around these four dimensions, with a total of 53 items initially identified.

### Ethical statements

**Ethics approval.** The relevant guidelines and regulations were followed in the Declaration of Helsinki. The study has been reported and approved by the Ethics Committee for Liaoning Normal University on 8th October 2024(Ethics approval number: LL2024165).

**Informed consent.** All participants voluntarily participated and signed informed consent forms. Moreover, all participants are adults. The anonymity of the participants was always guaranteed, and the information obtained was kept in total confidentiality and used only for research purposes.

**Consent for publication.** Consent for publication was obtained from all individual participants included in the study.

## Delphi method

**Expert selection.** In selecting consulting experts, the principles of breadth, representativeness, and authority were followed, taking into account relevant professional fields, geographic distribution, and response efficiency. Experts with relevant experience and research interests in various disciplines related to college students' physical activity were chosen. Based on the consensus reached by the number of experts selected during the scale development and evaluation process, a sample size of 10–15 people is sufficient to obtain reliable results [17]. When the number of selected experts approaches 15, further increasing the number of experts has little effect on prediction accuracy [18].

Ultimately, 16 experts from China in related fields agreed to participate in the questionnaire survey, including 13 professors, 1 associate professor, and 2 lecturers. Two rounds of correspondence surveys were conducted between January and March 2025, primarily utilizing email for the distribution and collection of questionnaires.

## Expert authority

Upon calculation, the average coefficient of judgment ($Ca$) is 0.96, and the average coefficient of familiarity ($Cs$) is 0.85. Thus, the average coefficient of authority ($Cr$) is 0.90, exceeding 0.70, indicating that the expert authority meets the requirements.

## Degree of expert opinion coordination

After two rounds of correspondence, the final 61 retained items achieved an expert opinion coordination coefficient of 0.252 ($P < 0.000$). The coefficient of variation ($CV$) in experts' importance ratings ranged from 0.061 to 0.316 across items, with only two items (Item A13 and Item C12) exhibiting a $CV$ exceeding 0.25. This indicates a high degree of coordination in expert opinions.

## Expert opinion concentration

After two rounds of correspondence, the average importance scores assigned by experts to the 61 retained items ranged from 3.909 to 4.909, all exceeding 3.500. The full-score ratios ranged from 0.273 to 0.909, all exceeding 0.200, indicating a high degree of consensus among experts regarding the importance of each item.

## Dimension evaluation

The mean importance scores for the four dimensions—physical activity capacity, physical activity behavior, physical activity cognition, and emotional experience— were 4.93, 4.86, 4.86, and 4.71, respectively. The $CV$ ranged from 0.05 to 0.13, all below the commonly used standard of 0.25. Therefore, as evaluated by experts, all dimensions were retained。

## Item adjustments

Following the first round of expert consultation, items were modified, added, or deleted based on the ratings and suggestions. Overall adjustments focused on three directions: (1) Enhancing the accuracy of certain words and phrasing to avoid ambiguity and improve item answer ability; (2) Strengthening the connection between some items and "physical activity", "daily life", and "active health" to emphasize the theme of physical literacy;(3) Addressing differences and synergies among sections to enhance complementarity while minimizing redundancy. Recommendations for "deletion" or "merging" items were handled cautiously. Only 5items were deleted or merged, carefully weighing expert suggestions while ensuring compliance with the coefficient of variation requirements. For deletion suggestions made by only a few experts, we primarily revised the wording to enhance scientific rigor, integrating qualitative and subsequent quantitative methods wherever possible. Ultimately, 24items were modified, 13 were added, and 5 were deleted. Thus, after the first round of revisions, the total number of items increased from 53 to 61.

 

For the second round of expert consultation, to facilitate scoring and enhance review efficiency, individualized explanations of modifications were provided to each expert. Justifications were also offered for recommendations not adopted after careful consideration. Simultaneously, the survey form clearly identified modified, added, and deleted items, appended the original items, and highlighted modified items to facilitate targeted expert scoring and enhance overall response validity. Following the second round of expert consultation, expert ratings on item importance largely stabilized, requiring only descriptive revisions to specific items. For instance, the original statement "Even minor collisions or falls (e.g., tripping, minor bumps) could cause me to fracture a bone" was modified to "I am prone to fracturing bones from minor collisions."

After two rounds of expert consultation and revision, the final version comprises four dimensions: physical activity capacity, physical activity behavior, physical activity cognition, and emotional experience, totaling 61 items(Table 1).

## Questionnaire survey method

This study primarily employed two rounds of questionnaire surveys. The first round focused on item screening and confirmatory factor analysis, while the second round conducted a large-scale survey. Participants for this survey were recruited between 2 April and 28 April 2025.

## First distribution

Based on the findings from the preliminary Delphi expert consultation, the 61 identified items were distributed via questionnaire for quantitative evaluation of the items and assessment of the scale's performance. Inclusion criteria for research subjects were: non-physical education majors, physically and mentally healthy, and capable of participating in regular school physical education programs. Considering economic, educational, and cultural differences, this study employed a combination of convenience sampling and stratified sampling to maximize coverage across China's four major economic regions and various types of higher education institutions. All questionnaires in this section were distributed online, yielding 1,468 responses. Invalid samples—including those lacking informed consent, failing lie-detector questions, or completing the survey too quickly—were excluded, resulting in 1,306 valid responses (88.96% response rate). Sample characteristics are detailed in Table 2 below. The 1,306 valid samples were randomly divided into two parts: Data A ($n = 653$) and Data B ($n = 653$). Data A was used for exploratory factor analysis, with its sample size meeting the requirement of 5–10 times the number of items. Data B was employed for confirmatory factor analysis, meeting the minimum sample size requirement of 200 for structural equation modeling, supported by most scholars [19].

## Second Distribution

This questionnaire survey employed a combined strategy of convenience sampling and stratified sampling to investigate the physical literacy levels of Chinese college students. The inclusion criteria for research subjects were identical to those of the first questionnaire survey. A total of 3,289 questionnaires were collected. After data cleaning, 3,077 valid samples were obtained, yielding a valid response rate of 93.55%. In this round of investigation, reliability and validity testing showed that the overall *Cronbach's* α coefficient was 0.814, with all dimensions exceeding 0.7, indicating that this study possesses good reliability. The final sample information is presented in Table 3 below. The sample encompasses the primary types of institutions within China's higher education system, demonstrating a degree of representativeness. Overall, the research sample exhibits sufficient variability and coverage in terms of gender, grade, and household registration, etc., reflecting the structural characteristics of China's college student population to a certain extent. This provides a necessary sample foundation for subsequent analyses of population differences in physical literacy among Chinese college students.

This survey questionnaire consists of two parts. One part consists of personal information, covering details such as gender, grade, and household registration, etc. The other part employs the self-developed "Chinese College Student

**Table 1. Initial items.**

| NO. | Items |
| --- | --- |
| A1 | I have excessive abdominal fat. |
| A2 | I believe my weight is within a healthy range. |
| A3 | I am prone to fractures from minor collisions. |
| A4 | I am able to sustain moderate-intensity endurance exercise for over 30 minutes consecutively, such as long-distance running, swimming, or cycling. |
| A5 | I can engage in physical activity for extended periods without feeling fatigued. |
| A6 | I can effortlessly perform tasks requiring considerable strength, such as weightlifting and moving heavy objects. |
| A7 | Compared to my peers, I am able to complete more consecutive standard pull-ups (for men)/sit-ups (for women). |
| A8 | My standing long jump distance surpasses that of most of my classmates. |
| A9 | I run the 50 meters faster than most of my classmates. |
| A10 | I can touch my toes when performing a seated forward bend. |
| A11 | When exercising, my movements are coordinated and fluid. |
| A12 | I can maintain a one-legged stance with my eyes closed for longer than my classmates. |
| A13 | I can naturally maintain rhythm in my daily activities (such as walking, running, dancing, etc.) |
| A14 | I am proficient in performing the fundamental movements of at least one common sport (such as basketball, football, swimming, etc.). |
| A15 | I can apply the motor skills I have learnt to other physical activities. |
| A16 | I am able to adapt my athletic skills flexibly according to different sporting contexts, such as competitions and training sessions. |
| B1 | I engage in vigorous physical activity every week and experience shortness of breath during such activities. |
| B2 | At least three days a week, I engage in physical activity that is somewhat strenuous and causes a noticeable increase in my heart rate. |
| B3 | I engage in at least 30 minutes of light physical activity on three or more days each week (including physical activity from work, commuting, household chores, leisure activities, etc.). |
| B4 | On weekdays, I spend most of my waking hours sitting or lying down. |
| B5 | On my days off, I spend most of my waking hours sitting or lying down. |
| B6 | I spend a considerable amount of time sitting or lying down using my mobile phone or computer almost every day. |
| B7 | I shall proactively manage my personal weight through measures such as regular exercise and healthy eating. |
| B8 | When sitting for extended periods (such as during work, study, or watching television), I will get up and move around for 3–5 minutes every hour. |
| B9 | Over the past month, I have frequently felt fatigued during everyday physical activities such as walking, climbing stairs, and doing household chores. |
| B10 | My sleep quality has been excellent over the past month. |
| B11 | After engaging in physical activity, I make a more conscious effort to maintain a regular sleep schedule. |
| B12 | I have an exercise programme and plan tailored to my individual circumstances. |
| B13 | I shall take thorough precautions to prevent sports injuries, such as warming up, stretching, and wearing appropriate protective gear. |
| B14 | During physical activities, I shall monitor my physical condition and adjust or cease promptly to avoid injury. |
| C1 | I understand the health benefits of physical activity. |
| C2 | I know how to improve my health through physical activity. |
| C3 | I am aware of the physical activity guidelines recommended by relevant authoritative bodies. |

*(Continued)*

**Table 1.** (Continued)

| NO. | Items |
|---|---|
| C4 | I know what constitutes a healthy lifestyle. |
| C5 | I understand how to synergistically promote personal health through a healthy diet and regular physical activity. |
| C6 | I am aware of the detrimental effects of prolonged sitting on personal health, such as increasing the risk of cardiovascular disease and contributing to obesity. |
| C7 | I understand the specific health benefits of different physical activities, such as improving cardiovascular health, enhancing muscle strength, and increasing flexibility. |
| C8 | I understand that adults should engage in at least 150 minutes of moderate-intensity physical activity each week, such as brisk walking or cycling. |
| C9 | Participating in sports helps improve my social skills. |
| C10 | I have acquired the knowledge to avoid sports-related risks. |
| C11 | I have acquired the knowledge to handle sports accidents in emergencies. |
| C12 | I am aware of the scientific principles governing warm-up and cool-down exercises. |
| C13 | I know that when experiencing physical discomfort (such as fatigue, pain, dizziness, etc.), one should cease physical activity. |
| C14 | I have cultivated a lifelong commitment to physical fitness. |
| C15 | For me, regular participation in physical activity is essential. |
| C16 | I am willing to invest time and effort in physical exercise as part of my daily routine. |
| C17 | I am committed to making active participation in physical activity a sustained way of life. |
| C18 | I know how to utilise existing space for physical activity. |
| D1 | For me, engaging in physical activity on a regular basis is a source of pleasure, excitement or enjoyment. |
| D2 | I believe physical activity keeps me in better physical and mental condition. |
| D3 | I have forged friendships and gained a sense of belonging through participating in team sports. |
| D4 | Through physical activity, I am able to gain a sense of achievement and satisfaction by overcoming challenges. |
| D5 | I feel stressed or anxious during physical activities, particularly when faced with challenges or competition. |
| D6 | I often feel physically exhausted or lacking in motivation when engaging in physical activity. |
| D7 | I sometimes become completely absorbed in physical activity, forgetting everything around me. |
| D8 | I believe I can participate in and maintain physical activity on a regular basis. |
| D9 | I believe I possess sufficient skill and ability to participate in any sporting activity I wish to engage in. |
| D10 | I am satisfied with my level of physical fitness (such as cardiorespiratory fitness, muscular strength, and balance). |
| D11 | During physical activities, I am able to communicate and collaborate effectively with others. |
| D12 | Even when encountering setbacks during physical activities, I demonstrate resilience and self-assurance. |
| D13 | I am confident in my ability to perform well in physical activities. |

Physical Literacy Scale" to measure the physical literacy levels of Chinese college students. It encompasses four dimensions — physical activity capacity, physical activity behavior, physical activity cognition, and emotional experience—with a total of 32 items. College students are required to evaluate their own body composition, physical activity behavior, health cognition, attitudes, and other aspects. A 5-point Likert scale was used; higher scores indicate higher physical literacy levels among college students.

**Table 2. Sample description (I).**

| Variable | Category | Number(%) | Variable | Category | Number(%) |
|---|---|---|---|---|---|
| Gender | Man | 636(48.70%) | University District | Eastern | 513(39.28%) |
| | Woman | 670(51.30%) | | Central | 414(31.70%) |
| Household Registration | Urban | 713(54.59%) | | Western | 274(20.98%) |
| | Rural | 593(45.41%) | | Northeastern | 105(8.04%) |
| Grade | 2024Grade | 245(18.76%) | University Category | 985 Project, 211 Project, Double First-Class Initiative | 228(17.46%) |
| | 2023Grade | 277(21.21%) | | General Comprehensive University | 481(36.83%) |
| | 2022Grade | 263(20.14%) | | Academic Institutions | 376(28.80%) |
| | 2021Grade | 521(39.89%) | | Higher Vocational Education | 221(16.92%) |

**Table 3. Sample description (II).**

| Variable | Category | Number(%) | Variable | Category | Number(%) |
|---|---|---|---|---|---|
| Gender | Man | 1484(48.23%) | University District | Eastern | 1271(41.31%) |
| | Woman | 1593(51.77%) | | Central | 960(31.20%) |
| Household Registration | Urban | 1727(56.13%) | | Western | 578(18.78%) |
| | Rural | 1350(43.87%) | | Northeastern | 268(8.71%) |
| Grade | 2024Grade | 933(30.32%) | University Category | 985 Project, 211 Project, Double First-Class Initiative | 275(8.94%) |
| | 2023Grade | 803(26.10%) | | General Comprehensive University | 992(32.24%) |
| | 2022Grade | 785(25.51%) | | Academic Institutions | 859(27.92%) |
| | 2021Grade | 556(18.07%) | | Higher Vocational Education | 951(30.91%) |

## Mathematical statistics

All collected data were primarily organized using Excel, with missing values imputed using the mean. Descriptive statistics, item analysis, exploratory factor analysis, reliability and validity testing, independent samples t-tests, and analysis of variance were conducted using SPSS 27. Confirmatory factor analysis was performed using Amos 24. The significance level was set at $\alpha = 0.05$.

## Results

### Item analysis

**Test-retest reliability method.** Fifty respondents were selected for retesting. Significant correlation in item scores should be observed between the initial and retest surveys. Based on this criterion, 11 items warrant consideration for deletion: A12, A13, B7, B9, B10, B11, B14, C10, C11, C13, and D3.

**High-low group comparison method.** The high-low group comparison method is a statistical technique commonly used in the data screening process; typically, the sample groups comprising the top and bottom 27% are considered the optimal choice [20]. Respondents were ranked by total questionnaire scores. The top approximately 27% ($n = 364$) formed the high-score group, and the bottom approximately 27% (n = 366) formed the low-score group. Independent samples t-tests were conducted to compare score differences between groups for each item. Items failing to reach statistical significance were considered for deletion. In this study, only item A3 ($t = 1.046$, $P = 0.296$) showed no significant difference between groups and may be considered for deletion. All other items met the requirements.

**Internal item correlation method.** According to the Internal Item Correlation Method [21], using the correlation coefficient between each item and other items within its dimension as an indicator, if the correlation coefficient exceeds

0.9, it indicates duplicate content within that dimension, warranting deletion consideration, and items with extremely low reliability (<0.3) are also a sign that they should be removed from the provisional scale [22]. Simultaneously, within each dimension, if the number of internal items with a correlation coefficient below 0.2 for that item reaches over 50% (excluding the item itself and already deleted items), deletions should proceed sequentially based on proportional size. Calculations indicate a total of 10 items eligible for deletion: A2, A3, A10, B4, B6, B9, B10, C13, D5, and D6.

**Cronbach's alpha method.** Within each dimension, *Cronbach's α* coefficient is tested after deleting any single item. If deleting any item significantly improves reliability, that item is deemed unreasonable. In this study, four items fail to meet this criterion: A3, A10, D5, and D6.

## Summary of item screening results

Based on the statistical results above, 20 items were identified as potentially problematic. Items A3, A10, B9, B10, C13, D5, and D6 were repeatedly screened as failing to meet the item analysis criteria and were directly deleted. The remaining items were designated as pending for further consideration. At this stage, the total number of items decreased from 61 to 54.

## Validity testing

**Exploratory factor analysis.** An exploratory factor analysis was conducted on the 54 items using Data Set A to further refine the items while revealing their internal structure. The Kaiser-Meyer-Olkin (*KMO*) and Bartlett's sphericity test results were: KMO = 0.933 > 0.7, P < 0.001, indicating that the data were suitable for exploratory factor analysis.

Drawing on exploratory factor analysis methods commonly used in previous studies [23], factor rotation was performed using the maximum variance method, amd the following principles were applied: (1) Commonality must not be less than 0.35;(2) Factor loadings ≥ 0.4; (3) The Presence of cross-loadings; (4) Items assigned to certain factors were difficult to interpret. Items B7 and C9 had factor loadings below 0.4, items C3 and C11 exhibited cross-loadings, and items A13, B8, B11, C10, and C12 were assigned to other factors and were difficult to interpret. After optimizing the model, a 10-factor, 45-item model was obtained, explaining 61.604% of the cumulative variance (>60%).

Based on the aforementioned research findings and prior empirical distributions of factors, physical literacy can be further subdivided into a 10-dimensional structure, sequentially named as follows: (1) Factor 1 comprises 8 items primarily concerning fundamental athletic qualities such as strength, speed, and endurance, thus named "Athletic Qualities";(2) Factor 2 comprises 6 items, primarily concerning an individual's confidence in their abilities, behaviors, and performance, thus named "Confidence"; (3) Factor 3 comprises 5 items, primarily concerning an individual's cognitive attitudes and willingness toward physical activity, thus named "Attitude and Willingness"; (4) Factor 4 comprises 7 items, primarily concerning an individual's understanding of the relationship between physical activity and health, thus named "Health Cognition";(5) Factor 5 comprises 5 items, primarily concerning individual's emotions and feelings during physical activity, thus named "Emotions"; (6) Factor 6 comprises 3 items, primarily concerning individual's mastery of motor skills, thus named "Motor Skills"; (7) Factor 7 comprises 3 items, primarily concerning individual's sedentary and screen-based behaviors, thus named "Sedentary and Screen-Based Behaviors";(8) Factor 8 comprises 3 items, primarily concerning individual's risk- avoidance behaviors toward physical activity, thus named "Risk Avoidance Behavior"; (9) Factor 9 comprises 3 items, primarily concerning individual's physical activity behaviors in daily life, thus named "Physical Activity"; (10) Factor 10 comprises 2 items, primarily concerning individual's waist circumference and body weight, thus named "Body Composition".

**Confirmatory factor analysis.** Confirmatory factor analysis was conducted using Data Set B to further validate the internal structural validity of each dimension.

Physical activity capacity Dimension: After model refinement, four items (A5, A7, A8, A11) with relatively high Modification Indices (MI) and Expected Parameter Change (EPC) values were removed, resulting in a final 3-factor, 9-item model as shown below. Fitting indices: $χ^2/df = 2.604 < 3$, P < 0.001, indicating excellent fit; *RMSEA* = 0.050 < 0.08, indicating excellent fit; *GFI* = 0.979, *AGFI* = 0.961, *CFI* = 0.976, *IFI* = 0.976, all exceeding 0.9, demonstrating good model fit. Collectively, these results validate the 3-factor, 9-item model for the Physical activity capacity dimension (Fig 1)

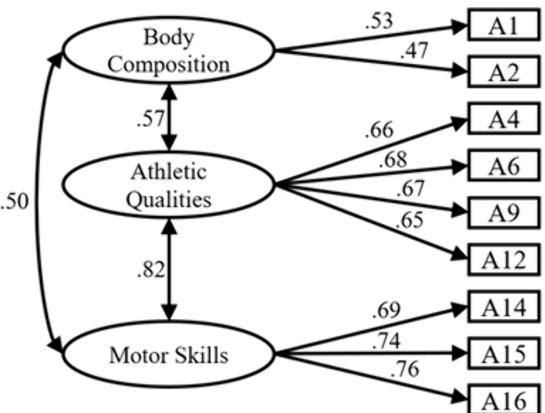

**Fig 1. Structural diagram of physical activity capacity.**

Physical Activity Behavior Dimension: Using Data Set B, confirmatory factor analysis was conducted to further validate the internal factor structure of this dimension. After removing B14 following MI correction, a 3-factor, 8-item model was ultimately obtained, as shown below. The fit indices are as follows: $\chi^2/df = 2.395 < 3$, $P < 0.001$, indicating excellent fit; $RMSEA = 0.046 < 0.08$, indicating excellent fit; $GFI = 0.985$, $AGFI = 0.968$, $CFI = 0.977$, $IFI = 0.977$, all exceeding 0.9, demonstrating good model fit. Collectively, these results confirm the validation of the 3-factor, 8-item model under the physical activity behavior dimension (Fig 2).

Physical Activity Cognitive Dimension: Using Data B for confirmatory factor analysis, we further validated the internal factor structure of this dimension. After MI correction, items C2, C7, C8, and C18 were removed, ultimately yielding a 2-factor, 8-item model as shown below. Fitting indices: $\chi^2/df = 1.912 < 3$, $P < 0.05$, indicating excellent fit; $RMSEA = 0.037 < 0.08$, indicating excellent fit; $GFI = 0.987$, $AGFI = 0.975$, $CFI = 0.989$, $IFI = 0.989$, all exceeding 0.9, demonstrating good model fit. Collectively, these results confirm the validity of the 2-factor, 8-item model within the physical activity cognition dimension (Fig 3).

Emotional Experience Dimension: Using Data Set B for confirmatory factor analysis further validated the internal factor structure of this dimension. After MI correction, items D3, D8, D9, and D10 were removed, ultimately yielding a 2-factor,

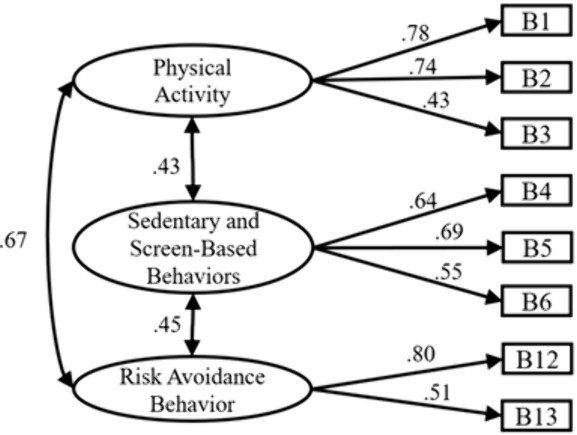

**Fig 2. Structural diagram of physical activity behavior.**

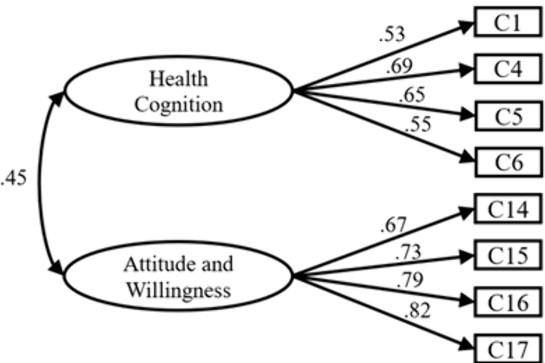

**Fig 3. Structure diagram of physical activity cognition.**

7-item model as shown below. Fitting indices: $\chi^2/df = 2.581 < 3$, $P < 0.01$, indicating ideal fit; $RMSEA = 0.049 < 0.08$, indicating excellent fit; $GFI = 0.985$, $AGFI = 0.968$, $CFI = 0.985$, $IFI = 0.985$, all exceeding 0.9, demonstrating good model fit. Collectively, these results confirm the validation of the 2-factor, 7-item model under the emotional experience dimension (Fig 4).

Combining all items and dimensions results in a comprehensive higher-order model, as shown in Fig 5. Following comprehensive model validation, the factor load ranges for all items are between 0.46 and 0.82. Key fit indices indicate: $\chi^2/df = 4.159 < 5$, $P < 0.01$, indicating excellent fit; $RMSEA = 0.049 < 0.08$, indicating excellent fit; $GFI = 0.909$, $CFI = 0.906$, $IFI = 0.906$, all exceeding 0.9. The results indicate a good fit. The combined results confirm the validity of this model. Therefore, it can be concluded that the physical literacy model comprises four primary factors—physical activity capacity, physical activity behavior, physical activity cognition, and affective experience—along with ten secondary factors. Specifically, the physical activity dimension encompasses body composition, athletic qualities, and motor skills; the physical activity behavior dimension includes physical activity, sedentary and screen-based behaviors, and risk-avoidance behaviors; the physical activity cognition dimension covers health cognition, attitudes, and willingness; and the emotional experience dimension comprises confidence and emotions.

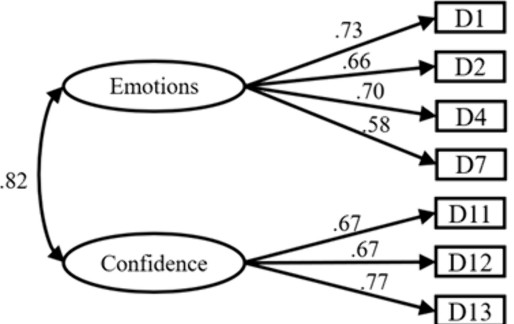

**Fig 4. Structure diagram of emotional experience.**

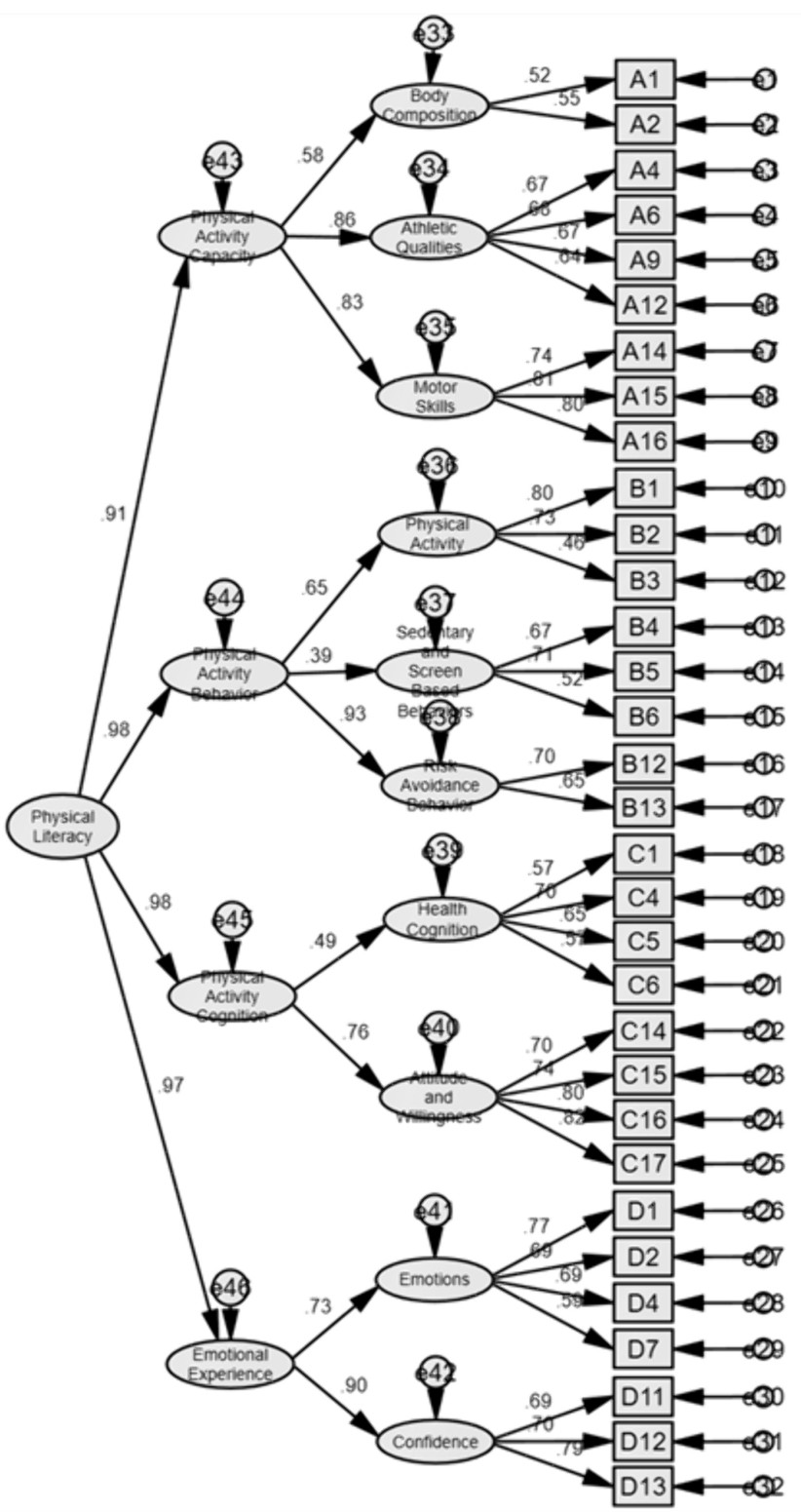

**Fig 5. Diagram of the complete model.**

### Reliability testing

Internal consistency, reliability, and test-retest reliability were employed. *Cronbach's* alpha coefficients for the four dimensions--physical activity capacity, physical activity behavior, physical activity cognition, and emotional experience-- ranged from 0.731 to 0.821, all exceeding 0.7, indicating good reliability of the scale. Retest reliability assessed the correlation between two measurement sets. Item correlation coefficients ranged from 0.512 to 0.772, all exceeding 0.5. This indicates a highly significant correlation between the two sets of data, suggesting good test-retest reliability.

In summary, after multiple rounds of screening (the items process is illustrated in Fig 6, the final College Student Physical Literacy Scale comprises 32 items across four dimensions: physical activity capacity, physical activity behavior, physical activity cognition, and emotional experience. This scale is suitable for measuring the physical literacy levels of Chinese college students.

### Differences in physical literacy among Chinese college students

**Overall differences in college students' physical literacy.** The findings of the population study on college students' physical literacy are presented in Table 4. Significant differences in physical literacy were observed across gender, household registration, grade, university type, and university location, indicating that these variables influence students'

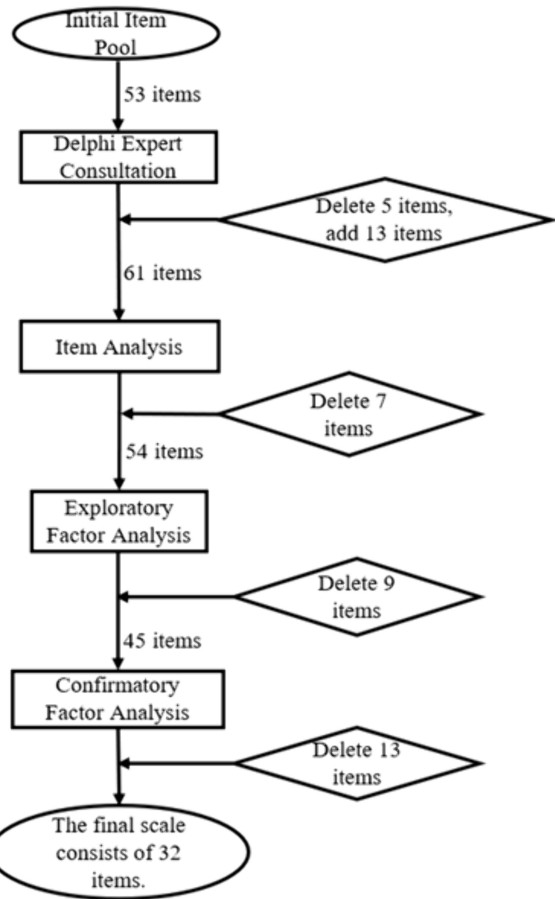

**Fig 6. Item screening process diagram.**

**Table 4. Analysis of physical literacy.**

| Variable | Category | M ± SD | t/F |
|---|---|---|---|
| Gender | Man | 3.73 ± 0.56 | 24.914*** |
| | Woman | 3.24 ± 0.55 | |
| Household Registration | Urban | 3.59 ± 0.59 | 11.464*** |
| | Rural | 3.34 ± 0.60 | |
| Grade | 2024Grade | 3.53 ± 0.60 | 30.811*** |
| | 2023Grade | 3.55 ± 0.60 | |
| | 2022Grade | 3.35 ± 0.57 | |
| | 2021Grade | 3.26 ± 0.62 | |
| University Category | 985 Project, 211 Project, Double First-Class Initiative | 3.55 ± 0.60 | 17.305*** |
| | General Comprehensive University | 3.50 ± 0.60 | |
| | College | 3.32 ± 0.60 | |
| | Higher Vocational Education | 3.47 ± 0.60 | |
| University District | Eastern | 3.49 ± 0.61 | 4.957** |
| | Central | 3.46 ± 0.60 | |
| | Western | 3.43 ± 0.57 | |
| | Northeastern | 3.60 ± 0.61 | |

Note: ** $P<0.01$, *** $P<0.001$.

physical literacy levels. Specifically: Urban students demonstrated significantly higher physical literacy than rural students; Students in the 2024 cohort exhibited higher physical literacy levels than those in the 2022 and 2021 cohorts, indicating a declining trend in physical literacy during the university years; Differences in physical literacy were observed across university categories, with students from college-type institutions showing relatively lower levels, while those from 985, 211, and Double First-Class universities demonstrated relatively higher levels; Physical literacy levels also vary by university location, with students in western regions exhibiting relatively lower levels and those in northeastern regions showing relatively higher levels, reflecting an overall pattern of higher levels in the east and lower levels in the west. These findings reveal that physical literacy among college students is not uniformly distributed but is systematically influenced by factors such as gender, geography, and educational resources, reflecting the projection of uneven socioeconomic development onto physical literacy. This necessitates that universities, while addressing internal disparities within student populations through targeted interventions to promote educational equity, also establish long-term, four-year holistic development mechanisms to counteract the decline in physical literacy across grades. Enhancing students' physical literacy must be regarded as a core component of institutional physical education objectives, thereby genuinely cultivating robust, well-rounded new generations for the modern era.

### Differences in internal dimensions of college students' physical literacy

Building on overall physical literacy disparities, the study now shifts focus to its core dimensions—physical activity capacity, behavior, cognition, and emotional experience—to examine internal heterogeneity and subgroup patterns, supporting targeted, multi-tiered educational interventions.

As shown in Table 5, the internal dimensional differences in college students' physical literacy align with the overall findings on physical literacy. Significant differences persist across all four dimensions based on gender, household registration, grade, university type, and university location. Male students consistently scored significantly higher than female students across all dimensions, with the largest gap observed in physical activity behavior. Urban students outperformed rural students significantly in all dimensions, with the most pronounced difference in physical activity capacity. Grade level

**Table 5. Comparisons across dimensions of physical literacy.**

| Variable | Category | Physical activity capacity | Physical activity behavior | Physical Activity Cognition | Emotional Experience |
|---|---|---|---|---|---|
| Gender | Man | 3.65±0.69 | 3.23±0.71 | 4.09±0.53 | 4.01±0.62 |
| | Woman | 3.01±0.70 | 2.54±0.66 | 3.83±0.59 | 3.64±0.74 |
| | *t/F* | 25.106*** | 28.049*** | 12.701*** | 15.143*** |
| Household Registration | Urban | 3.46±0.74 | 3.00±0.78 | 4.03±0.54 | 3.92±0.67 |
| | Rural | 3.15±0.76 | 2.72±0.72 | 3.86±0.61 | 3.69±0.74 |
| | *t/F* | 11.152*** | 10.225*** | 7.719*** | 9.154*** |
| Grade | 2024Grade | 3.42±0.77 | 2.93±0.76 | 3.98±0.60 | 3.86±0.69 |
| | 2023Grade | 3.39±0.77 | 3.00±0.76 | 4.01±0.53 | 3.87±0.71 |
| | 2022Grade | 3.16±0.72 | 2.67±0.72 | 3.88±0.57 | 3.75±0.70 |
| | 2021Grade | 3.10±0.74 | 2.60±0.78 | 3.80±0.60 | 3.62±0.76 |
| | *t/F* | 25.118*** | 41.904*** | 14.365*** | 12.428*** |
| University Category | 985 Project, 211 Project, Double First-Class Initiative | 3.43±0.81 | 2.89±0.78 | 4.05±0.53 | 3.88±0.66 |
| | General Comprehensive University | 3.35±0.75 | 2.92±0.78 | 3.96±0.57 | 3.84±0.70 |
| | College | 3.14±0.73 | 2.69±0.72 | 3.83±0.60 | 3.68±0.76 |
| | Higher Vocational Education | 3.27±0.73 | 2.97±0.69 | 3.95±0.60 | 3.76±0.76 |
| | *t/F* | 16.198*** | 13.504*** | 15.254*** | 10.012*** |
| University District | Eastern | 3.34±0.76 | 2.88±0.81 | 3.97±0.56 | 3.81±0.69 |
| | Central | 3.31±0.74 | 2.86±0.70 | 3.93±0.58 | 3.79±0.73 |
| | Western | 3.25±0.71 | 2.81±0.72 | 3.92±0.57 | 3.81±0.69 |
| | Northeastern | 3.42±0.95 | 2.99±0.92 | 4.07±0.57 | 4.00±0.76 |
| | *t/F* | 3.263* | 3.415* | 5.392** | 4.980** |

Note: $*P < 0.05, **P < 0.01, ***P < 0.001$

showed significant differences across all dimensions, with physical activity capacity, physical activity behavior, physical activity cognition, and emotional experience all exhibiting a declining trend as grade level increased, most markedly in physical activity behavior. Students from different university categories demonstrated significant differences in all dimensions of physical literacy. In physical activity capacity, physical activity cognition, and emotional experience dimensions, students from 985, 211, and Double First-Class universities generally exhibited relatively higher levels, while students from vocational colleges showed the highest levels in physical activity behavior and the lowest levels across all dimensions among college students. Double First-Class universities showed relatively higher levels. In physical activity behavior, vocational college students demonstrated the highest levels, while college students scored lowest across all four dimensions. Differences also emerged based on university location: students from western regions exhibited relatively lower physical literacy levels, while those from northeastern regions scored relatively higher across all dimensions.

## Discussion

### Gender differences: A dual product of biological and social construction

Male college students demonstrated significantly higher physical literacy levels than female students, a finding requiring interpretation at both biological and social levels. Biologically, males typically possess advantages in traditional physical fitness indicators such as muscle strength and explosive power, directly reflected in their scores on the "physical activity capacity" dimension. However, the more critical factor lies in the social construction dimension. Traditional gender

role perceptions have forged a strong association between masculinity and sports participation, while females are often directed toward static, aesthetic activities. This results in female students generally having fewer opportunities, lower frequency, and shorter duration of participation in moderate-to-high intensity physical activities compared to males, particularly in competitive and confrontational team sports. Additionally, female students may be more prone to body image concerns such as objectification anxiety and shyness during physical activities [24], leading to lower scores on the emotional experience dimension. Sociocultural cues may also cause some girls to underestimate their physical capabilities, fostering a fixed mindset of being unathletic that undermines confidence and motivation to participate. These gender disparities in physical activity emerge as early as childhood. Numerous studies show boys consistently score higher than girls in physical literacy, physical competence, and motor skills [25,26]. However, experimental studies also indicate that women often demonstrate more significant improvements in knowledge, understanding, motivation, and confidence compared to men. This may stem from their heightened responsiveness to social support [27,28]. In other words, appropriate interventions by educational institutions can bridge gender-related gaps in physical literacy among college students, thereby altering the pattern of lower physical literacy levels among female students.

Therefore, gender issues constitute an indispensable factor in university interventions targeting physical literacy levels. Particular attention should be paid to regulating female students' emotional and cognitive responses, leveraging their biological characteristics while remaining vigilant against structural gendered cultural oppression and reshaping more comprehensive perceptions of gender and the body.

## Urban-rural disparities reflect unequal resource allocation and cultural accumulation

The physical literacy levels of urban college students are significantly higher than those of rural students, reflecting the direct impact of China's urban-rural dual structure on health outcomes. First, micro-system resources are scarce. Disparities in resource endowments between urban and rural schools result in stark differences in their capacity to develop competitive sports and inclusive physical education [29]. Rural schools suffer from severe shortages in sports infrastructure and specialized physical education faculty, making it difficult to provide high-quality, diverse physical education. This places rural students at a disadvantage from the outset in developing foundational physical activity skills. Second, support from the meso-system is weak. Rural families have relatively limited economic and cultural capital, making it difficult to afford extracurricular sports training, purchase specialized athletic equipment, or accompany children to participate in sports activities. This limits the breadth and depth of their sports socialization. Finally, macro-level cultural influences persist. Traditional rural educational philosophies often prioritize "academic achievement above all," with the college entrance exam serving as a key pathway for "children from humble backgrounds to achieve greatness." This mindset suppresses students' physical activity cognition and engagement. Naturally, such economic disparities in physical activity are prevalent globally. Communities with relatively weaker economic conditions exhibit significantly reduced student physical activity levels [30]. These economic disparities affect students' physical activity not only through material resources for sports but also via neighborhood social cohesion, neighborhood safety perceptions, and the built environment [31,32]. In other words, students' physical literacy is influenced not only by tangible resources but also by intangible cultural atmospheres.

Therefore, for students from diverse backgrounds, efforts must be made to address capacity gaps and establish equitable starting points. Support systems should be designed to accommodate varying levels and needs, with particular emphasis on daily foundational support rather than performance levels, fostering an inclusive rather than competitive daily atmosphere.

## Grade-level disparities reveal systemic compression eroding individual agency

The trend of declining physical literacy with increasing grade levels warrants particular vigilance. This reveals structural contradictions within the current higher education environment. On the one hand, curriculum development lags. University

 

physical education courses are typically concentrated in lower grades, leaving upperclassmen without mandatory, organized physical activities or guidance on healthy exercise. Their physical activity habits become highly dependent on established routines and self-discipline, which often prove insufficient under pressure. On the other hand, the dual pressures of academic demands and employment expectations squeeze out healthy habits among students [33,34]. As students advance through higher grades, they face immense academic pressure alongside multiple demands such as graduate school entrance exams, professional certification tests, internships, and job hunting. Personal time becomes severely compressed, making physical activity the first luxury to be sacrificed. This also indicates that in the face of powerful institutional and structural pressures, in reality, individuals gradually lose their capacity for autonomous choice. Additionally, this may relate to shifts in social support systems. For instance, research indicates that social support plays a significant role in shaping individual health behaviors [35]. Among college students, lower-year cohorts engage in more collective activities centered around classes and dormitories, whereas upper-year students tend toward individualization and dispersion. The dissolution of existing exercise-based social circles reduces social support, weakening the persistence of physical activity.

Therefore, mitigating the grade-related decline effect hinges on overcoming time barriers and reshaping exercise habits. It is particularly crucial to capitalize on the formative window during the early years. By refining the curriculum system and optimizing environmental support, institutions can reshape individuals' health awareness. Integrating physical activity habits into daily life through subtle, gradual means fosters a stable sports lifestyle, enabling students to withstand future life pressures and competing demands.

### Interschool and regional disparities reflect policy and cultural convergence

Differences between university categories and regions result from the combined effects of macro-level educational resource allocation and institutional positioning. Leading institutions typically possess superior athletic facilities, a wider array of sports course options, and higher-caliber faculty, providing students with a more advantageous physical environment. They also maintain more comprehensive student health management systems, including psychological and behavioral counseling, as well as education on exercise habits and health concepts, which are more conducive to cultivating students' physical literacy. Comprehensive universities with dedicated sports colleges, in particular, find their campus athletic culture more readily translated into a healthy and positive social environment, subtly enhancing students' cognitive and emotional experiences. Similarly, regional development disparities manifest as greater economic prosperity in eastern regions, leading to higher university funding and richer community sports cultures. This creates a favorable ecosystem linking campus and community activities. In contrast, Western universities face relatively limited resources, and regional cultural openness may influence students' acceptance of emerging and diverse physical activity forms. Thus, differences in college students' physical literacy stem from the combined effects of economic, educational, and cultural factors.

Although inter-institutional and regional disparities are guided by macro-level policies, university administrators can exercise autonomy to implement meso-level management of student physical literacy through incentive mechanisms, foundational health services, and institutional safeguards. Within constrained higher education sports environments, this approach maximizes the long-term mechanisms for holistic talent development.

### Conclusion

The physical literacy model for Chinese college students proposed in this study demonstrates satisfactory fit indices, indicating strong validity. Furthermore, the internal consistency reliability and test-retest reliability across all dimensions are acceptable, confirming the scale's reliability. In summary, the 32-item Chinese College Student Physical Literacy Scale, which comprises four dimensions—physical activity capacity, physical activity behavior, physical activity cognition, and emotional experience—is suitable for assessing the physical literacy levels of Chinese college students. The survey reveals distinct patterns in the physical literacy of this population. Overall, male students, urban residents, and

undergraduates in lower grade levels exhibit higher levels of physical literacy. Additionally, differences exist across institutional types and geographic regions, with universities in eastern China outperforming those in western China. These findings collectively indicate a multidimensional gap in physical literacy development. The disparities in physical literacy among Chinese college students stem from individual, social, and environmental factors. Specifically, gaps related to gender, urban-rural residence, grade level, and geographic region reflect broader social, policy, and cultural influences. To address these issues, universities should adopt ecosystem-based interventions aimed at improving student fitness, promoting holistic development, and enhancing the quality of education.

## Supporting information

**S1 File. Delphi-data only.**
(XLSX)

**S2 File. Data.**
(XLSX)

## Acknowledgments

We sincerely thank everyone who assisted in the successful completion of this study.

## Author contributions

**Conceptualization:** Rongjing Ni, Ying Yu.

**Data curation:** Rongjing Ni.

**Formal analysis:** Rongjing Ni.

**Investigation:** Rongjing Ni.

**Project administration:** Ying Yu.

**Supervision:** Ying Yu.

**Writing – original draft:** Rongjing Ni.

**Writing – review & editing:** Rongjing Ni, Ying Yu.

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
