## [Decision Letter · Decision Letter 0]

16 Feb 2026

PONE-D-25-65782Research on the Development and Differentiation of the Physical Literacy Scale for Chinese College StudentsPLOS One

Dear Dr. Yu,

Thank you for submitting your manuscript to PLOS ONE. After careful consideration, we feel that it has merit but does not fully meet PLOS ONE’s publication criteria as it currently stands. Therefore, we invite you to submit a revised version of the manuscript that addresses the points raised during the review process.

Please note that we have only been able to secure a single reviewer to assess your manuscript. We are issuing a decision on your manuscript at this point to prevent further delays in the evaluation of your manuscript. Please be aware that the editor who handles your revised manuscript might find it necessary to invite additional reviewers to assess this work once the revised manuscript is submitted. However, we will aim to proceed on the basis of this single review if possible. 

We look forward to receiving your revised manuscript.

Kind regards,

Jennifer Tucker, PhD

Staff Editor

PLOS One

Journal Requirements:

3. We note that you have indicated that there are restrictions to data sharing for this study. PLOS only allows data to be available upon request if there are legal or ethical restrictions on sharing data publicly. For more information on unacceptable data access restrictions, please see http://journals.plos.org/plosone/s/data-availability#loc-unacceptable-data-access-restrictions

Reviewers' comments:

Reviewer's Responses to Questions

**Comments to the Author**

1. Is the manuscript technically sound, and do the data support the conclusions?

Reviewer #1: Partly

2. Has the statistical analysis been performed appropriately and rigorously? 

Reviewer #1: Yes

3. Have the authors made all data underlying the findings in their manuscript fully available?

Reviewer #1: No

4. Is the manuscript presented in an intelligible fashion and written in standard English?

Reviewer #1: No

5. Review Comments to the Author

Reviewer #1: This manuscript presents the development and validation of a Physical Literacy Scale for Chinese college students using Delphi methodology, item analysis, and exploratory factor analysis (EFA). The topic is relevant and contributes to the growing field of physical literacy assessment in higher education populations.

The study demonstrates methodological effort and large sample size (n = 3,077), which strengthens statistical robustness. Other strengths of this manuscript include a systematic scale development process and a good statistical screening steps. However, several areas require clarification and refinement, particularly in methodological transparency, statistical justification and in the theoretical framing.

Detailed suggestions can be obtained from the attached reviewer report.

6. PLOS authors have the option to publish the peer review history of their article (what does this mean?). If published, this will include your full peer review and any attached files.

Reviewer #1: No

---

## [Author Response · Author response to Decision Letter 1]

30 Mar 2026

Point-to-point response

I am deeply grateful to the reviewers for your valuable suggestions on my research. I would also like to thank the editor for their assistance throughout this process. I have carefully considered these comments and revised the manuscript accordingly. Below are my responses to each suggestion:

1. The author should add specific statistical values in the abstract.

Some relatively important research findings have been supplemented in the abstract.

2. Ensure all abbreviations are defined at first use (e.g., KMO, Ca, Cr, Cs).

Based on the modification suggestions you have proposed, the full forms of these abbreviations have been provided, such as in lines 146, 147, 281, etc.

3. The author is advised to define and clarify the theoretical model that underpins the four-dimension structure. Mention if the framework is adapted from existing models or newly conceptualized by the authors.

It has been Supplement in lines 110-112 that the four-dimensional framework for physical literacy referenced the Physical Literacy Evaluation Index System Part 1: Adults and Part 2: Adolescents, rather than being newly constructed. This indicator system model was public released by an official Chinese institution and holds a certain degree of authority within China，It has been widely recognized and referenced by numerous scholars in China.

4. The manuscript requires clear justification of the expert selection criteria. Explain if there was a sample size calculation formula or a predefined consensus threshold in selecting the 16 experts.

The additions and modifications have been made according to your suggestions, the criteria for expert selection—including professional fields, regional breadth, and corresponding efficiency, etc—have been outlined in Lines 134–140. The selection of expert numbers also references consensus standards in the field of scale development and includes citations.

5. Multiple screening methods were used in the study namely, Internal item correlation, Cronbach’s Alpha, and EFA. It is good if the author can include the sequence of deletion steps, preferably in a flowchart. Justify the use of PCA instead of the common factor analysis.

The entry screening process has been visualized as a flowchart per your suggestion. The image is displayed on page 23, with a reference provided at line 365.

6. Authors should justify the cutoffs of 0.9 and 0.2 with proper citation and to explain the reason a 50% threshold was chosen.

We have incorporated your suggestions by adding reference annotations to the numerical standards (0.9, 0.2, 50%, etc.) within the Internal Item Correlation Method on line 259, referencing the commonly accepted consensus standards for this method in China.

7. Provide information on how the extracted 10 factor structure related to the originally mentioned four dimensions.

We have added relevant explanations in lines 349-356. These 10 factors belong to the second-order factors.

8. The ethical statement is provided appropriately. However, clarify how the anonymity was maintained in the dataset. Confirm whether participation was voluntary and whether any compensation was provided.

We have detailed the ethics statement under the Methods section as requested in the email, and provided supplementary explanations regarding anonymity, voluntariness, and other details, primarily in lines 117-125.

9. The restriction on data availability restricted the reviewer from being able to check on the results and the presentation.

Regarding data acquisition, we regret that we are unable to present the complete dataset at this time. As this portion of the data is also relevant to another ongoing research project within our group, we cannot disclose it in its entirety. Once all related studies are completed, we will be able to provide the full dataset.

10. Minor language editing is required.

The language in the manuscript has been polished.

We have carefully implemented the revisions you suggested above. Should you have any further valuable suggestions, please feel free to share them. We will continue to spare no effort in refining this work. Thank you once again for your invaluable assistance with this research.

---

## [Editor Report · Decision Letter 1]

9 Apr 2026

PONE-D-25-65782R1Research on the Development and Differentiation of the Physical Literacy Scale for Chinese College StudentsPLOS One

Dear Dr. Yu,

Thank you for submitting your manuscript to PLOS ONE. After careful consideration, we feel that it has merit but does not fully meet PLOS ONE’s publication criteria as it currently stands. Therefore, we invite you to submit a revised version of the manuscript that addresses the points raised during the review process.

We look forward to receiving your revised manuscript.

Kind regards,

Othman A. Alfuqaha, Ph.D.

Academic Editor

PLOS One

Journal Requirements:

**Additional Editor Comments:**

Dear Authors,

Thank you for your detailed point-by-point response and for the revisions made to the manuscript. I read your paper with great pleasure. Before proceeding with the final decision, I would appreciate your clarification on the following points to ensure that all reviewer concerns have been fully and rigorously addressed:

- Some methodological justifications (e.g., cut-off values, PCA vs. common factor analysis, and expert selection criteria) would benefit from stronger support through international references, not only local or regional standards.

- The explanation of the relationship between the 10 extracted factors and the original four dimensions requires clearer conceptual justification to ensure theoretical coherence.

- Regarding data availability, while your justification is understood, please include a formal data availability statement in the manuscript clarifying access conditions.

- Minor language improvements were mentioned; however, I notice some problems in whole manuscript. Please make sure that all are fine.

- Please confirm that all ethical considerations (anonymity, voluntariness, and consent procedures) are explicitly and clearly stated within the Methods section.

-Finally, I would like to see a Figure in AMOS program with all items and subdminesions.

Once these points are clearly addressed, the manuscript will be in a stronger position for final evaluation.

Thank you for your efforts and cooperation.

Best regards,

Dr. Alfuqaha

---

## [Author Response · Author response to Decision Letter 2]

16 Apr 2026

Point-to-Point Response

Dear Academic Editor/Reviewer,

Thank you very much for your recognition of my research. We would also like to express my sincere gratitude for your valuable suggestions on improving the manuscript, which have been instrumental in refining its content. We have carefully addressed each of the proposed revisions, and my detailed responses are provided below.

1- Some methodological justifications (e.g., cut-off values, PCA vs. common factor analysis, and expert selection criteria) would benefit from stronger support through international references, not only local or regional standards.

Following your valuable suggestions, we have incorporated relevant references from the international literature into the sections concerning cut-off values, factor analysis, and expert selection. These additions can be found in lines 144, 258–260, 268–272, and 293.

2- The explanation of the relationship between the 10 extracted factors and the original four dimensions requires clearer conceptual justification to ensure theoretical coherence.

We have further clarified the relationships between the four main dimensions and the ten extracted factors in lines 116–121 and 357–364. These ten factors represent lower-order constructs derived from the original four dimensions. Additionally, to present these relationships more intuitively, we have added Figure 5.

3- Regarding data availability, while your justification is understood, please include a formal data availability statement in the manuscript clarifying access conditions.

We appreciate you bringing this important point to our attention. In response, we have now added a formal data availability statement, which can be found on line 558.

4- Minor language improvements were mentioned; however, I notice some problems in whole manuscript. Please make sure that all are fine.

We have carefully implemented the revisions you requested, including adjustments to grammar, formatting, and wording.

5- Please confirm that all ethical considerations (anonymity, voluntariness, and consent procedures) are explicitly and clearly stated within the Methods section.

Thank you very much for raising this important issue. In response, we have now clearly outlined the relevant information regarding “anonymity, voluntariness, and consent procedures” in lines 128–130.

6-Finally, I would like to see a Figure in AMOS program with all items and subdminesions.

As you recommended, we have now added a complete AMOS model diagram （Figure5）to better illustrate the relationships among factors and items at each level, and its description begins at line 352.

We greatly appreciate the constructive feedback you have provided throughout the review process. In the revised manuscript, we have carefully addressed all of the comments and concerns raised. We believe that the changes made have significantly strengthened the paper. Should any further revisions be required, please do not hesitate to let us know. Thank you again for your time and expertise.

---

## [Editor Report · Decision Letter 2]

21 Apr 2026

Research on the Development and Differentiation of the Physical Literacy Scale for Chinese College Students

PONE-D-25-65782R2

Dear Dr. Ying Yu,

We’re pleased to inform you that your manuscript has been judged scientifically suitable for publication and will be formally accepted for publication once it meets all outstanding technical requirements.

Kind regards,

Othman A. Alfuqaha, Ph.D.

Academic Editor

PLOS One

Additional Editor Comments (optional):

Dear Authors,

Thank you for submitting your manuscript to the journal and for your careful revisions throughout the review process.

After a thorough evaluation of the reviewers’ comments and your responses, I am pleased to inform you that your manuscript has been accepted for publication.

Dr. Alfuqaha
---

## [Editor Report · Acceptance letter]

PONE-D-25-65782R2

PLOS One

Dear Dr. Yu,

I'm pleased to inform you that your manuscript has been deemed suitable for publication in PLOS One. Congratulations! Your manuscript is now being handed over to our production team.

Kind regards,

on behalf of

Dr. Othman A. Alfuqaha

Academic Editor

PLOS One